# Degree-Aware Graph Neural Network Quantization

**DOI:** 10.3390/e25111510

**Published:** 2023-11-02

**Authors:** Ziqin Fan, Xi Jin

**Affiliations:** Institute of Microelectronics, Department of Physics, University of Science and Technology of China, No. 96 Jinzhai Road, Hefei 230026, China; fzq@mail.ustc.edu.cn

**Keywords:** network quantization, graph neural network, graph data

## Abstract

In this paper, we investigate the problem of graph neural network quantization. Despite the great success on convolutional neural networks, directly applying current network quantization approaches to graph neural networks faces two challenges. First, the fixed-scale parameter in the current methods cannot flexibly fit diverse tasks and network architectures. Second, the variations of node degree in a graph leads to uneven responses, limiting the accuracy of the quantizer. To address these two challenges, we introduce learnable scale parameters that can be optimized jointly with the graph networks. In addition, we propose degree-aware normalization to process nodes with different degrees. Experiments on different tasks, baselines, and datasets demonstrate the superiority of our method against previous state-of-the-art ones.

## 1. Introduction

Different from regular data like images and videos, graph data are a special type of non-Euclidean irregular data, which cannot be directly processed by convolutional neural networks (CNNs). To remedy this, graph neural networks (GNNs) are developed to handle these irregularly structured data and are widely applied in applications like social networks [1], natural science [2,3], knowledge graphs [4], data mining [5] and recommendation systems [6,7]. Although GNNs are commonly shallower than CNNs with fewer parameters, their computational cost are tightly related to the input graph size. Considering that the graph size ranges from hundreds of nodes to billions of nodes, the high computational cost of GNNs becomes one of its major obstacle in real-world scenarios and hinders potential applications on resource-limited devices.

To improve the efficiency of GNNs, numerous network compression techniques successfully applied in CNNs have attracted increasing interests, including low-rank factorization, network pruning, network quantization, and knowledge distillation. Among these techniques, network quantization aims to represent the full-bit values in the network with low-bit ones and employ efficient integer arithmetic instead of expensive floating point arithmetic. As a result, the memory consumption and the computational cost of the quantized networks are significantly reduced without changing the architecture. Considering these advantages, this technique shows great potential for GNN acceleration.

Despite being successfully applied in CNNs, directly extending network quantization to GNNs may suffer severe performance drop [8]. Due to the high variance of node degree in a graph (ranging from one to hundreds or thousands), the magnitudes of the response for different nodes vary significantly in the GNN. This large range pose great challenge to existing network quantization methods and limits their accuracy. To address this issue, Tailor et al. [8] proposed Degree-Quant, which firstly considers node degree during network quantization and produces promising improvements. However, a heuristic scale parameter is used in Degree-Quant, which cannot flexibly adapt to diverse graph data. In addition, Degree-Quant relies on explicitly calculating the node degree to adaptively process nodes of varing degrees, which is quite time consuming.

To address the aforementioned issues, in this paper, we propose a degree-aware quantization for GNNs. Specifically, we introduce learnable scale parameters and optimize them together with the network parameters during training, allowing the quantizer to adapt to diverse graph data. In addition, we propose a simple yet effective degree-aware normalization method to normalize the response of nodes with different degrees to a common range. Our method is generic and compatible with different GNN architectures and tasks. The contributions of this paper can be summarized as follows:We develop a degree-aware quantization method for GNNs. Our method leverages learnable scale parameters to achieve flexible adaption to various graph data and employs a degree-aware normalization to avoid the adverse effect of varying node degrees to network quantization.We successfully apply our quantization method to typical GNN architectures and representative tasks (including node classification, graph classification, and graph regression). Extensive experiments validate the effectiveness of our designs and demonstrate the superiority of our method against previous approaches.

## 2. Related Work

In this section, we first briefly review several major works about graph neural network acceleration techniques. Then, we further discuss recent network quantization approaches that are closely related to our work.

### 2.1. Acceleration of Graph Neural Networks

Due to the capability of processing non-Euclidean data, GNNs have gained broad application and extensive research attention in various fields, such as complex systems [9] and social networks [10]. However, due to the high computational complexity of GNNs, these methods cannot be scaled to large-scale data. In order to handle large-scale graph data on resource-limited devices, numerous efforts have been made to accelerate graph neural networks, which can be roughly divided into graph-based and model-based approaches.

Graph-based methods aim at improving the speed of graph neural network by accelerating their graph operations. Early graph neural networks process full graphs and suffer redundant computational cost. To remedy this, techniques like graph sampling [11], graph sparsification [12,13], and graph partition [14] are widely studied to sample partial graphs to reduce the graph sizes, remove unimportant edges to increase the sparsity of the graphs, and divide the full graph into sub-graphs to obtain smaller ones. By reducing the computational consumption of graph operations, these methods achieve promising speedup.

Different from graph-based methods, model-based ones aim at accelerating graph neural networks by improving the efficiency of model operations. Specifically, Wu et al. and He et al. developed light graph neural network SGC [15] and lightGCN [16], which leverage lightweight network architectures and operation flows to achieve efficient training and inference. Meanwhile, several works employ generic network acceleration techniques like network pruning [17], knowledge distillation [18] and network quantization [19] for speedup. These methods do not require novel network designs and can improve the inference efficiency of existing graph neural networks, thereby attracting increasing interests. The technique of knowledge distillation involves pre-training a complex teacher model and then utilizing distillation loss to transfer the knowledge from the teacher model to a compact student model. This allows the student model to retain the knowledge of the teacher model. For example, KDGCN [18] proposed a knowledge distillation technique termed Local Structure Preservation module (LSP) to transfer knowledge for GCNs. Additionally, the KD-framework [20] presents an effective knowledge distillation framework that aims to achieve more efficient and interpretable predictions.

### 2.2. Network Quantization

Network quantization aims at representing full-bit floating-point numbers in the neural network with low-bit ones to reduce the memory and computational consumption. For example, by quantizing 32-bit weight and activation values to 4-bit ones, the model size is reduced to $\frac{1}{8}$, and the inference speed is significantly improved with the support of integer arithmetic, enabling efficient deployment of neural networks on FPGA platforms [21] or edge devices [22].

The feasibility and advantages of model quantization in traditional convolutional neural networks have been widely discussed. Networks such as BNN [23], TWN [24], and XNOR-Net [25] have been designed to quantize the weights to 1 or 2 bits, improving the inference speed at the cost of moderate performance drop. Inspired by the great success of network quantization in the area of convolutional neural networks [26], some studies extended this technique to graph convolutional neural networks. Specifically, Wang et al. [19] proposed Bi-GCN, which binarizes the input values and network parameters for speedup. In addition, to address the vanishing gradient issue during backpropagation caused by binarization, a new backpropagation method was designed for training. Tailor et al. [8] designed a quantization method tailored for GNNs, termed Degree-Quant. Specifically, they introduced a mask parameter to encourage nodes with higher degrees to retain their original accuracy, thereby avoiding the problem of large degree variations between different nodes. Despite promising performance, this approach introduces considerable memory and computational cost during training, which largely increases its training burden.

### 2.3. Self-Supervised Graph Representation Learning

In addition to high computational complexity, the annotation cost of graph data is also expensive and imposes challenges to GNNs. Motivated by the success of self-supervised learning in the field of natural language processing and computer vision, numerous efforts have been made to extend self-supervised learning to GNNs [27]. By contrasting similar nodes against dissimilar ones, discriminative representations can be learned from unlabeled data in an unsupervised manner, which can be transferred to downstream graph-based tasks to speedup the training phase [28,29]. Currently, self-supervised graph representation learning has drawn increasing interest.

## 3. Methodology

In this section, we first introduce the preliminaries. Then, we present our degree-aware quantization in detail.

### 3.1. Preliminaries

Network quantization aims at converting full-bit floating-point values in the network to low-bit ones to reduce the memory consumption and computational cost. Assuming that *x* represents a floating-point number and $x_{q}$ represents the quantized value, the quantization function can be defined as: (1)$$x_{q}=\mathrm{clamp}\left(\mathrm{round}\left(\frac{x}{S}+Z\right),\;\;Q_{min},\;\;Q_{max}\right).$$

Here, the clamp(·) function truncates the input number within the specified range, $Q_{max}$ and $Q_{min}$ are the maximum and minimum quantized values. For *N*-bit quantization, $Q_{max}=2^{N-1}-1,Q_{min}=-2^{N-1}$. Note that for unsigned numbers like activation values, $Q_{max}=2^{N}-1,Q_{min}=0$. *S* and *Z* are the scale parameter and the zero point, respectively. These two parameters can be calculated as: (2)$$S=\frac{q_{max}-q_{min}}{Q_{max}-Q_{min}},$$
(3)$$Z=\mathrm{clamp}(\mathrm{round}\left(Q_{max}-\frac{q_{max}}{S}\right),Q_{min},Q_{max}),$$
where $q_{max}$ and $q_{min}$ are the maximum and minimum values for floating-point numbers.

### 3.2. Degree-Aware Quantization

Our quantization framework with learnable scale parameters and degree-aware normalization is illustrated in Figure 1. Take a graph convolution as an example, the response from the previous layer *x* is first fed to the degree-aware normalization to normalize the values to a small certain range. Then, the normalized values and the convolutional kernel values are passed to the clip, scaling, and quantization steps, resulting in $x_{q}$ and $w_{q}$. Note that the scale parameters in the scaling step are differentiable and optimized by the task loss. With quantized values, graph convolution is conducted, and the result is then rescaled using the scale parameters. In this section, we first introduce the motivation. Then, we detail the degree-aware normalization and learnable scale parameter.

#### 3.2.1. Motivation

We first conduct experiments to study the values to be quantized in GNNs. As illustrated in [8], the responses of nodes with different degrees vary a lot. As shown in Figure 2, the nodes produce higher responses after the aggregation layer *x* as the node degree increases. Therefore, the values to be quantized span a wide range, which is difficult to be well covered by the quantizer. Meanwhile, it can be observed that the variance of the response values σ shares a similar trend as the response values *x*. This observation inspires us to use the variance values to normalize the response values. In this way, the responses are constrained to a small range (the red line), which facilitates the quantizer to quantize these values with low quantization errors for higher accuracy.

#### 3.2.2. Degree-Aware Normalization

In graph data, each node is connected to adjacent nodes, and this number of connected nodes is called the degree. Generally, the degree is highly uneven in a graph, ranging from one to hundreds or thousands. As discussed in Degree-Quant [8], the source of error in quantizing the graph convolutional neural network mainly comes from the aggregation phase. In this phase, nodes combine the feature information from its adjacent nodes in a permutation-agnostic manner. In the context of graph data, nodes with higher degrees collect more information from their adjacent nodes, resulting in a higher response after the aggregation phase. As a result, the range of full-bit responses for different nodes vary in a large range and introduce critical challenges for quantization.

The aggregation layer produces higher responses for nodes with higher degrees, shown in Figure 2. Meanwhile, the variance of response also increases linearly. Motivated by this observation, the aggregation responses are normalized by dividing their corresponding variances before being fed into the quantizer. This ensures that the input values produced by nodes with different degrees are constrained within a small certain range, facilitating the quantizer to reduce quantization errors for higher accuracy. Subsequently, we multiply the quantized results by the variance value again to ensure the range of the results.

#### 3.2.3. Learnable Scale Parameter

It is demonstrated in [30] that the scale parameter significantly affects the accuracy of the quantized networks. As discussed in the preliminary section, the scale parameters in current methods are usually pre-defined and fixed, which cannot flexibly fit various datasets, networks, and tasks. To remedy this, we develop learnable scale parameters to make them trainable during backpropagation. According to the quantization function in Equation (Equation 1), the gradient of the scale parameter is derived as: (4)$$\frac{\partial x_{q}}{\partial x}=\left\{\begin{array}{ccc} -\frac{x}{S}+round\left(\frac{x}{S}\right) & if & Q_{min}\leq \frac{x}{S}\leq Q_{max}\\ \;Q_{min} & if & \frac{x}{S}<Q_{min}\\ \;Q_{max} & if & \frac{x}{S}>Q_{max} \end{array}\right..$$

In our experiments, the scale parameters are initialized using *k* times standard deviation of the data in the first batch. By using trainable scale parameters, the quantizer can adaptively fit the distributions of full-bit values in diverse networks developed for various tasks and datasets.

## 4. Experimental Results

In this section, we first introduce the implementation details. Then, we compare our method with previous ones in terms of both accuracy and efficiency. Finally, we conduct experiments to validate the effectiveness of our method.

### 4.1. Implementation Details

#### 4.1.1. Datasets and Metrics

We conduct experiments on node classification, graph classification, and graph repression tasks. In our experiments, we employ the Cora dataset, the REDDIT-BINARY dataset, the MNIST-Superpixels dataset, and the ZINC dataset for both training and evaluation. The details of these datasets are described in Table 1. For classification tasks, we evaluate the model performance based on its accuracy on the test set. For regression tasks, we employ the mean absolute error (MAE) between the predicted values and actual labels as metrics.

The Cora dataset [31] contains a single graph representing a citation network, where each node corresponds to a research paper. The edges between nodes represent the citation relationships among the papers. The node features are binary indicators indicating the presence or absence of specific words in the corresponding papers. The task on the Cora dataset is to classify the nodes into their respective labels.The MNIST-Superpixels dataset is obtained from the MNIST dataset using SLIC [32]. Each graph is derived from a respective image, and each node represents a set of pixels or superpixels sharing perceptual similarities. This dataset is widely applied for the graph classification task.The REDDIT-BINARY dataset [33] consists of 2000 graphs corresponding to online discussions on the Reddit website. Each graph represents an online discussion thread, where nodes represent users and edges connect nodes if there has been a message response between the corresponding users. A graph is labeled according to whether it belongs to a question/answer-based community or a discussion-based community. This dataset is employed for the graph classification task.The ZINC dataset consists of molecules graphs, where each node represents an atom. The task involves graph regression [34], specifically predicting the constrained solubility based on the graph representation of the molecules.

#### 4.1.2. Baselines

In our experiments, three popular graph neural networks are employed as baselines, including the graph convolutional network (GCN), graph attention network (GAT), and graph isomorphism network (GIN). The setting of three baselines is presented in Table 2. In addition, as a message passing function [35] is the key process in GNNs, its formulation in different networks is detailed as follows.

Graph convolutional network (GCN) [36]:
(5)$$h_{v}^{t+1}=\sum_{w\in N(v)\cup \{v\}}\left(\frac{1}{\sqrt{d_{v}d_{w}}}Wh_{w}^{t}\right),$$
where $d_{v}$ represents the degree of node *v*.Graph attention network (GAT) [37]:In GAT, attention coefficients α are introduced and calculated based on task-specific query vectors and input information, allowing for higher weights on more valuable feature information. The message passing function is defined as follows:
(6)$$h_{v}^{t+1}=\alpha_{v,v}Wh_{v}^{t}+\sum_{w\in N(v)}\left(\alpha_{v,w}Wh_{w}^{t}\right).$$Here, the self-attention mechanism is utilized:
(7)$$e_{v,w}=LeakyReLU(a[Wh_{v}||Wh_{w}]),$$
where $e_{v,w}$ represents the significance of node *w* to node *v*, *a* is a single-layer feedforward neural network, and LeakyReLU is a non-linear activation function. Finally, the attention coefficients are obtained through a normalization layer:
(8)$$\alpha_{v,w}={softmax}_{w}\left(e_{v,w}\right)=\frac{exp(e_{v,w})}{\sum_{w\in N(v)}exp\left(e_{v,w}\right)}.$$Graph isomorphism network (GIN) [38]:GIN leverages the isomorphism property of graphs to complete diverse graph-related tasks. Its message passing function is defined as follows:
(9)$$h_{v}^{t+1}=f\left[\left(1+\epsilon \right)h_{v}^{t}+\sum_{w\in N(v)}h_{w}^{t}\right],$$
where *f* is a learnable injective function, such as a multi-layer perceptron (MLP), and ϵ is a learnable parameter.

#### 4.1.3. Experimental Setup

In our experiments, we utilized the Adam optimizer with $\beta_{1}=0.9$ and $\beta_{2}=0.999$ for training. For GCN on the Cora dataset and REDDIT-BINARY datasets, the batch size was set to 128. Meanwhile, for GCN on the MNIST and ZINC datasets, the batch size was set to 5. Other detailed hyperparameters, including the learning rate and the number of epochs, are presented in Table 3. For our method, an additional parameter *k* was employed to determine the initialization range of quantized data. We set k=3 for the Cora dataset and k=1 for the REDDIT-BINARY dataset.

### 4.2. Comparison with Previous Approaches

In this section, we compare our method with previous network quantization approaches on different tasks and datasets. Following previous evaluation protocol, we conducted over 10 runs on the MNIST-Superpixels and the ZINC dataset, and 10-fold cross validation on the REDDIT-BINARY dataset. We first report the accuracy results and then present the efficiency results.

#### 4.2.1. Accuracy

We first conducted the experiments following the implementation details in Section 4.1 and quantized both weight and activation values to INT8 and INT4, respectively. We opted for Degree-Quant (DQ) as the state-of-the-art method for comparison.

As shown in Table 4, our method produces comparable performance to the baseline model for both INT4 and INT8 quantizations. In addition, our method outperforms DQ by large margins, especially for INT4 quantization. For example, our method achieves an accuracy of 86.6% on Cora, surpassing DQ by over 16%. For the graph regression task, our method also produces lower errors than DQ (0.402 vs. 0.431). The superior performance of our method clearly demonstrates the effectiveness of our method.

#### 4.2.2. Efficiency

We then conducted experiments to study the training efficiency of our method. Specifically, we measured the average training time on the REDDIT-BINARY dataset using GIN as the baseline. Similarly, DQ was employed as the state-of-the-art method for comparison.

As shown in Table 5, our method achieves a significant efficiency improvement with a 12.9× speedup as compared to DQ. As discussed in Section 3.2.2, DQ relies on explicitly calculating the node degree and introduces considerable computational cost. In contrast, our method leverages the efficient degree-aware normalization with only negligible additional cost, thereby achieving significant speedup. This further demonstrate the high efficiency of our method.

### 4.3. Model Analyses

In this subsection, we conduct ablation experiments to study different components in our method, including the learnable scale parameter and degree-aware normalization.

#### 4.3.1. Learnable Scale Parameter

The learnable scale parameter enables the quantizer to flexibly adapt to various distributions of full-bit values in the network. To study its effectiveness, we replaced the learnable scale parameters with fixed ones for comparison. We conducted experiments on the Cora dataset. As shown in Table 6, the accuracy achieved by different graph networks suffers a notable drop when fixed-scale parameters are employed. This is because fixed-scale parameters cannot well fit diverse data in the dataset. This clearly validates the effectiveness of our learnable scale parameters.

#### 4.3.2. Initialization of Learnable Scale Parameters

As discussed in Section 3.2.2, the initialization of the learnable scale parameters is critical to the final performance. Consequently, we conduct experiments to study the effect of different initializations. Specifically, we set *k* to 0.5, 1, 3, and 5, and conduct experiments on the Cora dataset and the REDDIT-BINARY dataset for comparison. From Table 7, we can see that our method produces high accuracy on the Cora dataset and the REDDIT-BINARY dataset for k=3. Consequently, k=3 is used as the default setting of our method.

#### 4.3.3. Convergence of Learnable Scale Parameter

During training, our learnable scale parameters are optimized jointly with the network parameters. Therefore, we further conduct experiments to investigate their convergence during training. Specifically, we visualize the convergence of scale parameters in Figure 3. As we can see, the scale parameter is updated during training and gradually reaches convergence to fit the distributions of float values. This validates the effectiveness and flexibility of our learnable scale parameter.

#### 4.3.4. Degree-Aware Normalization

To address the issue that different nodes in a graph have different degrees, we employ degree-aware normalization to constrain the responses in a certain range. To validate its effectiveness, we remove this operation and compare its performance to our original method. The results are presented in Table 8. As we can see, our degree-aware normalization consistently introduces notable accuracy gains for different GNNs. For example, our degree-aware normalization produces an accuracy improvement of 4.6% on GCN for INT4 quantization. This demonstrates its effectiveness in handling the variation of node degrees in a graph.

## 5. Discussion

Our method shares a similar goal with Degree-Quant [8] to obtain low-bit GNNs for efficient inference. However, a heuristic scale parameter is used in Degree-Quant, which cannot flexibly adapt to diverse graph data. In addition, Degree-Quant relies on explicitly calculating the node degree to adaptively process nodes of varing degrees, which is quite time consuming. In contrast, our method leverages learnable scale parameters to achieve flexible adaption to various graph data and employs a degree-aware normalization to avoid the adverse effect of varying node degrees to network quantization. Moreover, one limitation of our method is that its accuracy suffers notable drops for bit widths lower than 4. In the future, we will conduct research to study binarized GNNs for further efficiency improvements.

## 6. Conclusions

In this paper, we propose a degree-aware network quantization method for graph neural networks. Specifically, we propose learnable scale parameters to fit various distributions of full-bit values in the network. In addition, we develop degree-aware normalization to handle the nodes with different degrees. Experiments demonstrate the effectiveness of our method against previous approaches on diverse tasks, datasets, and network architectures.

## Figures and Tables

**Figure 1 entropy-25-01510-f001:**
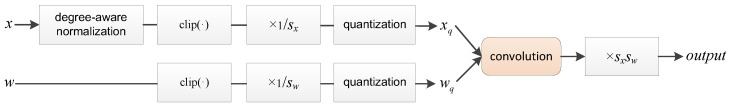
Quantization process with learnable scale and degree-aware normalization.

**Figure 2 entropy-25-01510-f002:**
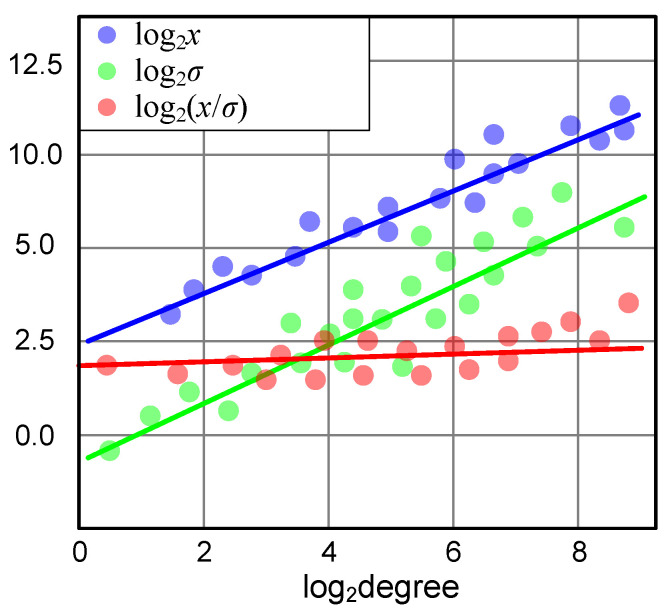
Analysis of values collected after aggregation at the final layer of FP32 GIN trained on Cora. *x* represents the response value, and σ represents the variance of *x*.

**Figure 3 entropy-25-01510-f003:**
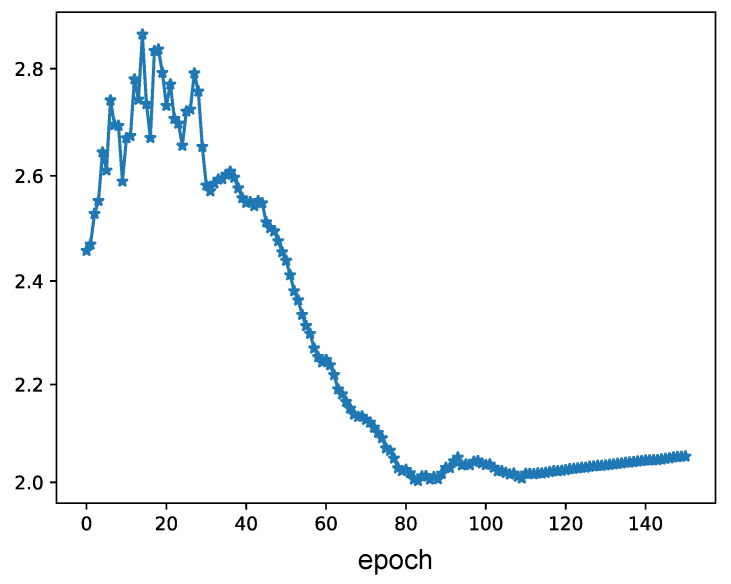
hlEvolution of the learnable scale parameter during training.

**Table 1 entropy-25-01510-t001:** Details of the datasets applied in this paper.

Dataset	Graphs	Nodes	Edges	Features	Labels	Task
Cora	1	2078	5278	1433	7	Node classification
MNIST	70K	40–75	565 (avg)	3	10	Graph classification
REDDIT	2K	430 (avg)	498 (avg)	1	2	Graph classification
ZINC	12K	9–37	50 (avg)	28	1	Graph regression

**Table 2 entropy-25-01510-t002:** Detailed parameters of the model architectures.

	Layers			Hidden Units		
	GCN	GAT	GIN	GCN	GAT	GIN
Cora	2	2	2	16	8	16
MNIST	4	4	4	146	19	110
REDDIT	–	–	5	–	–	64
ZINC	4	4	4	145	18	110

**Table 3 entropy-25-01510-t003:** Detailed hyperparameters of the experiments.

INT8	Learning Rate			Epoch		
	GCN	GAT	GIN	GCN	GAT	GIN
Cora	$10^{-2}$	$10^{-2}$	$10^{-2}$	750	500	400
MNIST	$5\times 10^{-5}$	$5\times 10^{-5}$	$5\times 10^{-4}$	1000	1000	1000
REDDIT	–	–	$10^{-2}$	–	–	200
ZINC	$5\times 10^{-5}$	$5\times 10^{-5}$	$5\times 10^{-4}$	1000	1000	1000
**INT4**	**Learning Rate**			**Epoch**		
	**GCN**	**GAT**	**GIN**	**GCN**	**GAT**	**GIN**
Cora	$10^{-2}$	$10^{-2}$	$10^{-3}$	750	500	400
MNIST	$5\times 10^{-5}$	$5\times 10^{-5}$	$5\times 10^{-4}$	1000	1000	1000
REDDIT	–	–	$10^{-2}$	–	–	200
ZINC	$5\times 10^{-5}$	$5\times 10^{-5}$	$5\times 10^{-4}$	1000	1000	1000

**Table 4 entropy-25-01510-t004:** Results produced by different methods on different datasets.

	Model	Cora (%)	MNIST (%)	REDDIT (%)	ZINC
Baseline (FP32)	GCN	81.4 ± 0.7	90.0 ± 0.2	–	0.469 ± 0.002
	GAT	83.1 ± 0.4	95.6 ± 0.1	–	0.463 ± 0.002
	GIN	77.6 ± 1.1	93.9 ± 0.6	92.2 ± 2.3	0.414 ± 0.009
DQ (INT8)	GCN	81.7 ± 0.7	90.9 ± 0.2	–	0.434 ± 0.009
	GAT	82.7 ± 0.7	95.8 ± 0.4	–	0.456 ± 0.005
	GIN	78.7 ± 1.4	96.6 ± 0.1	91.8 ± 2.3	0.357 ± 0.014
Ours (INT8)	GCN	83.0 ± 0.8	93.7 ± 0.1	–	0.371 ± 0.006
	GAT	90.4 ± 0.5	96.8 ± 0.2	–	0.361 ± 0.005
	GIN	86.8 ± 1.0	97.2 ± 0.1	91.9 ± 1.8	0.338 ± 0.007
DQ (INT4)	GCN	78.3 ± 1.7	84.4 ± 1.3	–	0.536 ± 0.011
	GAT	71.2 ± 2.9	93.1 ± 0.3	–	0.520 ± 0.021
	GIN	69.9 ± 3.4	95.5 ± 0.4	81.3 ± 4.4	0.431 ± 0.012
Ours (INT4)	GCN	80.8 ± 0.9	87.8 ± 0.2	–	0.455 ± 0.012
	GAT	88.4 ± 1.6	94.0 ± 0.1	–	0.410 ± 0.010
	GIN	86.6 ± 1.2	96.2 ± 0.2	89.2 ± 4.0	0.402 ± 0.011

**Table 5 entropy-25-01510-t005:** Training time (s) for different methods.

	DQ	Ours	Speedup
INT8	36.02	2.79	12.93×
INT4	35.98	2.78	12.94×

**Table 6 entropy-25-01510-t006:** Accuracy (%) achieved by models with and without learnable scale parameters.

INT8	GCN	GAT	GIN
w LSP	83.0	90.4	86.8
w/o LSP	57.2	85.2	85.2
**INT4**	**GCN**	**GAT**	**GIN**
w LSP	80.8	88.4	86.6
w/o LSP	48.2	85.8	86.0

**Table 7 entropy-25-01510-t007:** Accuracy achieved by models with different initialization values of learnable scale parameters.

INT8 (%)	Model	0.5	1	3	5
Cora	GCN	80.2	79.6	83.0	80.2
	GAT	30.6	26.6	90.4	89.4
	GIN	13.6	78.6	86.8	86.2
REDDIT	GIN	83.1	91.4	91.2	87.4
**INT4 (%)**	**Model**	**0.5**	**1**	**3**	**5**
Cora	GCN	80.2	79.4	80.8	80.6
	GAT	33.0	32.2	88.4	88.4
	GIN	30.8	64.6	86.6	85.4
REDDIT	GIN	80.8	89.2	71.6	55.4

**Table 8 entropy-25-01510-t008:** Accuracy (%) achieved by models with and without degree-aware quantization.

INT8	GCN	GAT	GIN
w DAN	83.0	90.4	86.8
w/o DAN	81.8	90.0	86.2
**INT4**	**GCN**	**GAT**	**GIN**
w DAN	80.8	88.4	86.6
w/o DAN	76.2	87.8	82.0

## Data Availability

Not applicable.

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
