# Peer review of "Degree-Aware Graph Neural Network Quantization"

_entropy, 2023, doi:10.3390/e25111510_

Round 1

Reviewer 1 Report

Comments and Suggestions for Authors

This paper proposes a degree-aware quantization method for graph neural networks (GNNs) to address the challenges of applying current network quantization approaches to GNNs. The proposed method leverages learnable scale parameters to achieve flexible adaptation to various graph data and employs degree-aware normalization to avoid the adverse effect of varying node degrees on network quantization.

The proposed degree-aware quantization method is generic and compatible with different GNN architectures and tasks, making it a versatile solution for graph neural network quantization. This flexibility allows the method to be applied to a wide range of real-world scenarios.

While the proposed degree-aware quantization method is a valuable contribution to the field of graph neural network quantization, further discussion of state-of-the-art graph representation learning frameworks enhanced by self-supervised learning techniques could enhance the paper.

Self-supervised learning techniques have shown great promise in improving the efficiency and accuracy of graph representation learning, and their integration with graph neural networks could lead to significant improvements in performance. Some recently developed self-supervised learning-enhanced graph neural networks include the following research investigations, which may need further discussions:

"Contrastive Graph Structure Learning via Information Bottleneck for Recommendation", NeurIPS'2022

"Self-supervised hypergraph transformer for recommender systems", KDD2022

One suggestion for improving the paper would be to include more detailed parameter setting information for the baseline methods used in the experiments. While the paper provides extensive experimental results demonstrating the effectiveness of the proposed degree-aware quantization method, it is not always clear how the baseline methods were implemented and what parameter settings were used.

Comments on the Quality of English Language

N.A

Reviewer 2 Report

Comments and Suggestions for Authors

In this paper, the authors analyze the problem of quantizing a GNN.

The topic considered by the authors is extremely interesting. The quantization methodology they propose appears convincing. The experiments performed are numerous and return satisfactory results. 

In my opinion there are two weaknesses in the paper that the authors should try to improve. In particular:

- The related work section needs to be supplemented with other papers that use or study GNNs through additional approaches. For example, there are recent attempts to study GNNs through complex network analysis or social network analysis. The authors should enrich the related work section by adding some of these approaches.

- Authors should add a "Discussion" section before the conclusions in which they illustrate the strengths, weaknesses, similarities and differences of their approach from those already proposed in the literature.

Comments on the Quality of English Language

The English appears good
